# The association between male involvement in institutional delivery and women's use of institutional delivery in Debre Tabor town, North West Ethiopia: Community based survey

**Kassanesh Melese Tessema[1], Kebadnew Mulatu Mihirete[2], Endalkachew Worku Mengesha[3], Azezu Asres Nigussie[4], Awoke Giletew Wondie[5]***

**1** Department of Reproductive Health, Debre Tabor Town Administration Health Center, Debre Tabor, Ethiopia, **2** Department of Epidemiology and Biostatics, Bahir Dar University, Bahir Dar, Ethiopia, **3** Department of Reproductive Health, Bahir Dar University, Bahir Dar, Ethiopia, **4** Department of Midwifery, Bahir Dar University, Bahir Dar, Ethiopia, **5** Department of Reproductive Health, Debre Tabor University, Debre Tabor, Ethiopia

* awokegiletew@yahoo.com

## Abstract

### Background

Maternal deaths remain high in Ethiopia mainly due to poor maternal health service utilization. Despite men are the chief decision-makers and economically dominant in Ethiopia, the impact of their involvement on maternal health services utilization is not clear. This study aimed to assess the association between male involvement and women's use of institutional delivery, and factors influencing male partners' involvement in institutional delivery.

### Methods

A community based cross-sectional study was conducted between March and May, 2019. A total of 477 married men who have children less than one year of age were interviewed. Face-to-face interviews using a pre-tested and structured questionnaire were used for data collection. Bivariate and multiple logistic regressions were carried out. SPSS version 23 was used for data analysis.

### Results

Overall 181 (37.9%) husbands/partners were involved in institutional delivery for the most recent child birth. Male partners involvement in institutional delivery was strongly associated with an increased odds of attending institutional delivery by spouse [AOR: 66.2, 95% CI: 24.8, 177.0]. Education [AOR: 0.33, 95% CI: 0.18–0.59], knowledge on maternal health [AOR: 1.67, 95% CI: 1.11–2.50], favourable attitude towards institutional delivery [AOR: 1.83, 95% CI: 1.23–2.71], and no fear while supporting spouse [AOR: 2.65, 95% CI: 1.28–5.50] were positively associated with male partners involvement in institutional delivery.

**Data Availability Statement:** All relevant data are within the manuscript and its Supporting information files.

**Funding:** The authors received no specific funding for this work.

**Competing interests:** The authors have decleared that no competing interests exist.

## Conclusion

Male partner's involvement in institutional delivery was inadequate. This study reported a significant beneficial impact of male involvement on maternal health through improved utilisation of institutional delivery. Therefore, maternal health interventions should target husbands as consumers of maternal health services, and healthcare/government policies that isolate or discourage men from having active engagement in maternal health should be improved.

## Introduction

Maternal and newborn health is still major concerns worldwide. Globally, approximately 800 women die every day due to pregnancy or childbirth related complications, and almost all maternal deaths (99%) occur in developing countries [1]. In Africa and South Asia, it is the leading cause of death for women of reproductive age [1]. The risk of a woman dying in sub-Saharan Africa as a result of pregnancy or childbirth is 1 in 39, as compared to 1 in 4,700 in industrialized countries [1, 2]. Ethiopia, as a sub-Saharan African country, is characterized by high maternal mortality (412 per 100,000 live births) [3], and direct obstetric complications account for 85% of maternal deaths [4]. These complications could be prevented or treated by skilled care during pregnancy, childbirth, and postpartum period [5]. Despite these proven interventions, maternal health service utilization in Ethiopian is one of the lowest in the world. Institutional delivery (26%) and PNC (Post Natal Care) service use (17%) has remained very low, while the ANC (Antenatal Care) coverage has shown an increment (27% in 2000 to 62% in 2016) but it is still below the desired [3]. This poor maternal health services utilization highlights the challenges to further improve maternal and child health [6].

Reduction of maternal mortality is a global priority and skilled care before, during and after childbirth is the key to save the lives of women and newborn babies [7, 8]. However, maternal deaths continuing to be significant in developing countries due to limited maternal health service utilization [9, 10]. This clearly indicate that the potential for improving maternal health through increasing utilisation of these key services has not been fully addressed, and remains one of the key areas for research and subsequent intervention [6]. Globally, the big challenge of the effort to improve maternal health is male attendance of skilled ANC and delivery care [11]. In sub-Saharan Africa absence of support from the husband is one of the main reasons for many pregnant women not to seek maternity services [12]. This indicates that the characteristics of health delivery system, economic and geographic accessibility, and socio-demographic factors are not the only factors affecting the uptake of maternal health care [13]. Therefore, the concept of male involvement in maternal health is now being promoted as a new essential element of World Health Organization (WHO) initiative for improving maternal and child health [14]. International Conference on Population and Development recommend that special efforts should be made for involving men in maternal and child health [15]. However, the term 'male involvement' is subjective and very multifaceted [16]. Most studies defined male involvement as the active participation of men in maternal health services/care (attending maternal health service and support/help); while others defined it as providing financial support for pregnancy and childbirth related expenses. The remaining studies defined male involvement as shared decision-making on maternal health with wife [17]. In this study male

involvement in institutional delivery is operationalized as the active participation of men in: accompanying for ANC, birth preparedness, encouraging for institutional delivery, and discussion with spouse, relatives and health professionals on place of delivery.

In most developing countries, the decision to seek care is usually made by male partners [18] and they are the chief providers which often determining women's access to economic resources [19]. Similarly in Ethiopia, women traditionally enjoy little independent decision making on most individual and family issues, including the option to choose whether to give birth in a health facility or seek the assistance of a trained provider [20]. High male involvement stemming from knowledge of delivery process results in positive perception towards skilled birth care and involvement manifested through such actions as physical and emotional and financial support which will lead to good maternal health outcomes [21]. These practice has implications for maternal health as it determines the nutritional status of women during pregnancy [22]; women's access to maternal health services since healthcare systems in most developing countries require out-of-pocket payments [23]; and women's chances of receiving emergency obstetrics care, which is vital in averting maternal mortality [22]. Even though this situation makes husbands a critical partner for the reduction of maternal mortality, in many sub-Saharan Africa countries including Ethiopia, maternal and child health is viewed as a woman's affair, and many studies on determinants of maternal health service utilization excluded men [18, 24].

Many studies have reported positive benefits of male involvement in maternal and child health including: increased uptake of maternal health care (ANC, institutional delivery, contraception use), decreased mother-to-child transmission of HIV, and improved maternal mental health [25–28]. However, other studies reported that it could occasionally lead to increased male dominance in decision-making, intimate partner violence, disruptions of family relationships and loss of economic support to women [28, 29].

Evidence suggests that male involvement may be beneficial to maternal health; however, the magnitude of the association is not clear and whether involving men in maternal health can improve maternal outcomes remains unclear [17, 30, 31]. There have also been speculations on possible negative impacts of male involvement in maternal health, hence it is necessary to explore the impact of male partner's involvement on the female partner's use of maternal health services in developing countries where the greatest burden of global maternal deaths and men's dominant roles in these countries have been shown to influence health outcomes. In Ethiopia where maternal death, males economic dominance and decision-making power remained high, attempts to examine male partner's involvement and its role on maternal health service utilization have been very limited, relying largely on socio-demographic and maternal characteristics [32, 33]. Thus, this study aimed to assess the association between male involvement in institutional delivery and women's use of institutional delivery, and identify factors influencing male involvement in institutional delivery in Debre Tabor town, Ethiopia.

## Methods

### Study design and area

A community based cross-sectional study was conducted between March and May, 2019 in Debre Tabor town, North West, Ethiopia. It is the city of South Gondar Zone in Amhara Regional State and located 667 kilometres from Addis Ababa, capital of Ethiopia. The town is divided in to six kebeles (small administrative units) with an estimated total population of 65530 in 2018.

## Population

The source population was all men in marital union with a women (legal, religious or traditional) having children less than one year of age and permanent residents of Debre Tabor town. All men in marital relationship having children less than one year of age and permanent residents of Debre Tabor town and selected during the study period were our study population.

## Sample size and sampling procedure

Firstly, the required sample size was calculated using single population proportion formula based on the prevalence (41.9%) of male involvement in institutional delivery in Ambo town assuming 95% confidence level of $Z_{a/2} = 1.96$, 5% of absolute precision, and 10% non-response rate [34] which yield sample size of 410. Moreover, sample sizes were calculated using factors associated with male involvement in institutional delivery, such as respondents' knowledge and perceived cost of health service [34, 35]. Finally, the maximum sample size of 486 was taken to address all objectives of the study (Table 1).

All 6 kebeles found in Debre Tabor town was included and a total of 2098 households that fulfil the inclusion criteria were selected. The list of households that fulfils the inclusion criteria (households with fathers in marital relationship having children less than one year of age and permanent resident) was obtained from health extension workers. Based on the obtained information, the sample size for each kebeles was proportionally allocated. The sampling interval for each kebeles was calculated by dividing the number of households that fulfil the inclusion criteria for allocated sample size in that kebeles. Finally, the study subjects were selected by systematic random sampling using every $k^{th}$ interval. The first father was selected by lottery method and subsequent fathers were included randomly based on the interval through house to house visit.

## Data collection

Data were collected by face to face interview technique using a structured and pre-tested questionnaire adapted from the survey tools developed by African Medical and Research Foundation (AMRF), Child and Reproductive Health Programme which is modified according to local context [36]. The questionnaire was prepared in English and translated into Amharic (the local language), and back into English to ensure consistency (S1 File). Six health extension worker and three BSc nurses was recruited as data collector and supervisors respectively. One day training was given for data collectors and supervisors on the techniques of data collection and purpose of the study prior to data collection.

As a data quality assurance measure, a pre-test was conducted on about 10% of the sample that has similar characteristics with the study population other than the sampled households in the study area before actual data collection period. The appropriateness and clarity of questions were checked and necessary modifications were made in terms of clarifying questions and vague terms, and removing items that were not aligned with the study objectives. Each item in the questionnaire was checked by the supervisor and the principal investigator on a

**Table 1. Sample size calculation based on factors affecting male partners' involvement in institutional delivery.**

| S.no | Factors | Assumptions | | | | | | | Sample size |
|------|---------|-------|---------|------|------|----------------------|--------------------------|---------------|-------------|
| | | Ratio | Power % | CI % | AOR | Involved among exposed | Involved among non-exposed | Design effect | |
| 1 | Knowledge of respondents | 2:1 | 80 | 95 | 2.9 | 79 | 56.4 | 1 | 173 |
| 2 | Perceived cost of health service | 2:1 | 80 | 95 | 1.79 | 45.3 | 31.6 | 1 | 441 |

daily basis to review the accuracy and completeness of variables. At the completion of data collection, all entered records were inspected for missing or incorrect values and data cleaning and coding was conducted before analysis by the principal investigator.

## Measurement

Male involvement in institutional delivery was our primary outcome variable computed based on responses to six items: accompanying spouse for ANC, participating in birth preparedness plan, encouraging spouse for institutional delivery, discussion with health professionals on the place of delivery, discussion with relatives and friends about the place of delivery. Therefore, men who responded "yes" to 3 or more items were categorized as "involved" while those who responded "yes" to less than 3 items were categorized as "not involved". Moreover, prior studies on male involvement in institutional delivery applied this approach of categorization [34–38]. Furthermore, place of delivery (institutional/home) for the last child was considered as another outcome variable to examine the association between male partners' involvement in institutional delivery and women's use of institutional delivery.

Knowledge on maternal health: Participants were asked 13 items on institutional delivery and signs of complication during pregnancy to assess their knowledge on maternal health. Men who responded correctly to 6 or more items were categorized as having "Good knowledge" while those who responded correctly to less than 6 items were categorized as "Poor knowledge" [39].

Attitude towards institutional delivery: Participants were asked to reflect their opinion on serious of six items concerning institutional delivery using likert scale with a score ranging from 1 = strongly disagree to 5 = strongly agree. Using the mean score as the cut-off point, above or equal to 3.0 was categorized as "favourable attitude" and below 3.0 was categorized as having "unfavourable attitude" [39].

## Data analysis

Data were entered into Epi-data version 3.1 and exported to Statistical Package for Social Science (SPSS) version 23 for analysis. Descriptive statistics such as frequency mean and standard deviation was computed to describe variables of the study. Bivariate and multiple logistic regression analyses were fitted to assess the existence of association between male partner's involvement and women's use of institutional delivery. Another model was fitted to identify factors associated with male partners' involvement in institutional delivery. For both models explanatory variables on bivariate logistic regression analysis with $p<0.2$ was entered into multivariable logistic regression analysis and a p-value of less than 0.05 was used to declare statistical significance.

## Ethics approval and consent to participate

Ethical clearance was obtained from Institutional Review Board of Bahirdar University (IRB). Official letter of cooperation was obtained from the respective administrators in the study area. Informed written consent was obtained from individual participants before data collection. Confidentiality was strictly maintained throughout the process.

## Results

### Socio-demographic characteristics

Out of 486 estimated study samples, 477 had a full response and included in the analysis making a response rate of 98%. The mean age of study participants were 33.8 years with Standard

Deviation (SD) of ±5.9 years; and the age range lies between 20 and 52 years. Regarding the level of education, 240 (50.3%) had diploma and above and the average monthly income of respondents were 3,241.9 Ethiopian Birr with SD of ± 1756 Birr (Table 2).

## Health service utilization and accessibility related characteristics

Two hundred ninety (60.8%) men reported that their spouse gave birth in health institutions for their most recent birth. Majority of them (60.4%) made decision on place of delivery with their spouse jointly (Table 3).

## Knowledge on maternal health

More than one third (34.6%) of respondents were knowledgeable, while others (65.4%) have poor knowledge on maternal health. Majority of respondents know the signs of pregnancy complications (87.4%) and the importance of institutional delivery (69.4%) (S1 Table).

## Attitude towards institutional delivery

In this study about 234 (49.1%) male partners had favourable attitude towards institutional delivery, while the rest 243 (50.9%) had unfavourable attitude (S2 Table).

**Table 2. Socio-demographic characteristics of study participants in Debre Tabor town, North-West Ethiopia, 2019 (n = 477).**

| Variables | Frequency | Percent |
|---|---|---|
| **Age (years)** | | |
| 20–29 | 170 | 35.6 |
| 30–39 | 249 | 52.2 |
| $\geq$ 40 | 58 | 12.2 |
| **Religion** | | |
| Orthodox | 398 | 83.4 |
| Muslim | 52 | 10.9 |
| Protestant | 27 | 5.7 |
| **Ethnicity** | | |
| Amhara | 459 | 96.2 |
| Oromo | 10 | 2.1 |
| Tigrie | 8 | 1.7 |
| **Educational status** | | |
| Diploma & above | 240 | 50.3 |
| Secondary education | 110 | 23.1 |
| Primary education | 86 | 18.0 |
| No formal education | 41 | 9.0 |
| **Monthly income (ETB)** | | |
| < = 1000 | 35 | 7.3 |
| 1000–2500 | 162 | 34.0 |
| 2501–5000 | 222 | 46.5 |
| > = 5000 | 58 | 12.2 |
| **Occupation** | | |
| Civil servant | 246 | 51.6 |
| Private employ | 36 | 7.5 |
| Business men | 124 | 26.0 |
| Unemployed | 71 | 14.9 |

ETB = Ethiopian Birr

**Table 3. Health service utilization and accessibility related characteristics of respondents in Debre Tabor town, North-West Ethiopia, 2019 (n = 477).**

| Variables | Frequency | Percent |
|---|---|---|
| **Frequency of ANC visit by spouse (for the youngest child)** | | |
| Zero | 282 | 59.1 |
| One | 116 | 24.3 |
| Two | 62 | 13.0 |
| Three | 11 | 2.3 |
| Four | 6 | 1.2 |
| **Place of delivery (for the youngest child)** | | |
| Health institution | 290 | 60.8 |
| Home | 187 | 39.2 |
| **Decision on place of delivery** | | |
| Jointly with spouse | 288 | 60.4 |
| Spouse | 156 | 32.7 |
| Male partner | 30 | 6.3 |
| Other | 3 | 0.6 |
| **Distance from health facility (reachable in 30 minutes)** | | |
| Yes | 364 | 76.3 |
| No | 94 | 19.7 |
| I am not sure | 19 | 4.0 |
| **Perceived cost of health service** | | |
| Absolutely free | 381 | 79.8 |
| Affordable | 74 | 15.5 |
| Expensive | 22 | 4.6 |

## Socio-cultural barriers for male involvement

This study identified several socio-cultural barriers that hinder male partners' involvement in supporting their spouses to access institutional delivery. One third (33.5%) of respondents believed that child birth is a natural phenomenon that should not be given much attention and about 30% reported that it's not our culture to discuss with wife about place of delivery (S1 Fig).

## Male partners involvement in institutional delivery

Overall 181 (37.9%) husbands/partners were involved in institutional delivery for the most recent child birth (95% CI = 34%–42%). Discussion with relatives on the place of delivery (77%) and encouraging spouse for institutional delivery (76.5%) were the most common activities while discussion with health worker on place of delivery (36.9%) was the least common activity of involvement (Table 4).

## The association between male involvement and women's use of institutional delivery

Table 5 shows the association between male involvement in institutional delivery and women's use of institutional delivery in Debre Tabor town. The bivariate logistic regression analysis identified age, education, occupation, income, fear of being seen by others, knowledge and attitude of male partners, and male partners involvement as a significant variable. However, in multiple logistic regression analysis, male involvement and knowledge on maternal health was significantly associated with women's use of institutional delivery for the most recent birth.

**Table 4. Male partners involvement in institutional delivery in Debre Tabor town, North-West Ethiopia, 2019 (n = 477).**

| Variables | Frequency | Percent |
|---|---|---|
| **Accompany spouse for ANC follow-up** | | |
| Yes | 195 | 40.9 |
| No | 282 | 59.1 |
| **Birth preparedness support by male partners** | | |
| Yes | 330 | 69.2 |
| No | 147 | 30.8 |
| **Encourages institutional delivery** | | |
| Yes | 166 | 76.5 |
| No | 51 | 23.5 |
| **Discuss with health provider about the place of delivery** | | |
| Yes | 176 | 36.9 |
| No | 301 | 63.1 |
| **Discuss with relatives about the place of delivery** | | |
| Yes | 187 | 77 |
| No | 56 | 23 |
| **Discuss with friends about the place of delivery** | | |
| Yes | 165 | 67.9 |
| No | 78 | 32.1 |

Male partners involvement in institutional delivery appeared to be strongly associated with an increased odds of attending institutional delivery by spouse [AOR: 66.2, 95% CI: 24.8, 177.0] as compared to those men not involved. Moreover, women with knowledgeable male partners had about 6 times [AOR: 5.8, 95% CI: 3.18, 10.78] more odds of using institutional delivery when compared to women whose partner had poor knowledge on maternal health (Table 5).

## Factors associated with male involvement in institutional delivery

Table 6 shows factors associated with male partner's involvement in institutional delivery at Debre Tabor town. Age, education, occupation, income, fear of being seen by others, knowledge and attitude of male partners towards maternal health were significant variables on bivariate logistic regression. After adjusting for the effect of confounding variables in the multivariate analysis, educational status, knowledge on maternal health, attitude towards

**Table 5. The association between male involvement and women's use of institutional delivery in Debre Tabor town, 2019 (n = 477).**

| Variable | Institutional delivery | | COR(95% CI) | AOR(95% CI) | P-value |
|---|---|---|---|---|---|
| | **Yes** | **No** | | | |
| **Knowledge on maternal health care** | | | | | |
| Poor | 156(53.8) | 156(83.4) | 1 | 1 | 0.0000 |
| Good | 134(46.2) | 31(16.6) | 4.32(2.76–6.77) | 5.854(3.18–10.78) | |
| **Male partner involved in institutional delivery** | | | | | |
| No | 114(39.3) | 182(97.3) | 1 | 1 | 0.0000 |
| Yes | 176(60.7) | 5(2.7) | 56.2(22.4–140.1) | 66.26(24.8–177.0) | |

COR = Crude odd ratio, AOR = Adjusted odd ratio, CI = Confidence interval

**Table 6. Factors associated with male partner's involvement in institutional delivery at Debre Tabor town North-West Ethiopia, 2019 (n = 477).**

| Variable | Male involvement | | COR(95% CI) | AOR(95% CI) | P-value |
|---|---|---|---|---|---|
| | Yes (n = 181) | No (n = 296) | | | |
| **Education** | | | | | |
| Diploma and above | 116(64.1) | 122(41.2) | 1 | 1 | 1 |
| Secondary education | 30(16.6) | 80(27.1) | 0.39(0.24–0.64) | 0.44 (0.27–0.73) | 0.001*** |
| Primary education | 19(10.5) | 67(22.6) | 0.30(0.17–0.53) | 0.33 (0.18–0.59) | <0.001*** |
| No formal education | 16(8.8) | 27(9.1) | 0.62(0.32–1.22) | 0.62 (0.31–1.23) | 0.171 |
| **Knowledge on maternal health** | | | | | |
| Good | 79(44) | 86(29.1) | 1.89(1.29–2.78) | 1.67(1.11–2.50) | 0.014* |
| Poor | 102(56) | 210(70.9) | 1 | 1 | |
| **Attitude on institutional delivery** | | | | | |
| Favourable | 112(61.9) | 131(44.3) | 2.04(1.40–2.98) | 1.83(1.23–2.71) | 0.003** |
| Unfavourable | 69 (38.1) | 165(55.7) | 1 | 1 | |
| **Fear of being seen by others** | | | | | |
| Yes | 24(13.3) | 14(4.7.) | 1 | 1 | |
| No | 157(86.7) | 282(95.3) | 3.08(1.55–6.12) | 2.65(1.28–5.50) | 0.009** |

COR crudes odds ratio, AOR adjusted odds ratio,

* Significant at P<0.02;

** Significant at P<0.01;

*** Significant at P≤0.001

institutional delivery and socio-cultural barriers (fear of being seen by others) were significantly associated with male's involvement in institutional delivery.

Male partners who attend primary [AOR: 0.33, 95% CI: 0.18–0.59] and secondary education [AOR: 0.44, 95% CI: 0.27–0.73] had lower odds of involvement in institutional delivery as compared to those partners who had diploma and above.

Male partners with good knowledge on maternal health [AOR: 1.67, 95% CI: 1.11–2.50] and favourable attitude towards institutional delivery [AOR: 1.83, 95% CI: 1.23–2.71] had a higher odds of involvement in institutional delivery as compared to those partners who had poor knowledge and unfavourable attitude. Furthermore, male partners who did not fear others while supporting spouse had more than 2 times [AOR: 2.65, 95% CI: 1.28–5.50] odds of involvement in institutional delivery than those who fears while supporting spouse (Table 6).

## Discussion

This study investigated the association between male involvement in institutional delivery and women's use of institutional delivery, and identified factors associated with male involvement in institutional delivery among fathers who have children less than one year of age in Debre Tabor town, Ethiopia. In this study male involvement in institutional delivery was considered as the active participation of men in: accompanying spouse for ANC, birth preparedness, encouraging for institutional delivery and discussion with spouse, relatives and health professionals on place of delivery. Based on this we found that 37.9% of male partners were involved in at least half of total six activities of involvement for institutional delivery and studies which defined male involvement similarly reported closer level of involvement in Lemo woreda (38.2%) [35] and Ambo town (41.9%) [34], Ethiopia. This low level of involvement could result from low knowledge (34.6%), unfavourable attitude towards institutional delivery (50.9%), and socio-cultural barriers reported by this study. Generally, this inadequate male involvement

in a country like Ethiopia with poor maternal health services utilization highlights the challenges to further improve maternal and child health.

Regarding the specific activities of involvement, the current and previous studies shows that substantial proportion of male partners were involved in accompanying spouse for ANC and supporting birth preparedness, suggesting similar pattern of involvement by males partners across different setting in Ethiopia. In contrary, discussion with relatives on place of delivery was the most common activity of involvement in this study (77%) while it is the least common activity of involvement in Lemo woreda (15%) [35] and Ambo town (34.8%) [34]. This might be due to socio-cultural variation and rural setting of Lemo woreda which implies that priority should be given for improving awareness of male partner's on institutional delivery through community based health education in these settings.

This study examined the role of male partner involvement in institutional delivery as a determinant factor for female partners' utilization of institutional delivery from father's perspective. We found a statistically significant beneficial impact of male involvement on maternal health through improved utilisation of institutional delivery. Women whose husband involved in institutional delivery had more odds of attending institutional delivery than those whose husband were not involved; which implies the need to consider men as part of the solution and also to improve healthcare/government policies that isolate/discourage men from having active engagement in maternal health programs [23]. It is consistent with the findings of several studies in sub-Saharan Africa and elsewhere [17, 40–43], which suggested male partners' involvement in maternal health care during pregnancy has benefits on maternal health care services access and utilization. Other studies also have linked male partners' attendance to ANC with increased maternal health service utilization [17, 42]. This could be explained by the fact that the active involvement of male partners makes them more aware of the significance of maternal health care services which in turn makes them more likely to encourage and support their wives to use them [17, 40, 44]. In a developing country setting, this acquired knowledge could also translate into the husbands' grant of permission and provision of resources for accessing maternal services such as financial support for maternal health care services. On the other hand, the observed (60.4%) joint decision making with spouse on place of delivery in this study may contribute for the above strong association between male involvement and women's use of institutional delivery as shared decision-making on maternal health with wife is considered one features of male involvement.

Furthermore, the current study investigated factors affecting male partners' involvement in institutional delivery. We found that education, knowledge and attitude on maternal health and socio-cultural barriers have a significant influence on male involvement. Male partners' involvement in institutional delivery was shown to increases with increase in educational level of male partners, which reinforces the significant role that education plays in improving utilization of maternal health service. This finding is similar to the studies carried out in in Ethiopia [34, 45], Keyna [46] and Nigeria [47]. This is due to the fact that education leads to better health awareness, and may contribute for timely health care seeking and economic capability required.

This study showed that male partners who had good knowledge on maternal heath were more likely involved in institutional delivery than those who had poor knowledge which is consistent with other studies conducted in Lemo woreda [35], Ambo town [34] and Mareka woreda, southern, Ethiopia [45]. Those with good knowledge may understand well the possible birth complications; so that they encourage their spouses to give birth in health institutions. Similarly, male involvement was significantly higher among men's who showed favourable attitude towards maternal health service utilization. This was in line with other

studies conducted in Ethiopia and Nigeria [35, 48]. The fact that husbands' positive perception on benefit of maternity care might lead to higher level male partners' participation. These findings point to the important roles of partner's knowledge and attitude in influencing their involvement in institutional delivery which suggest a need to recognize husbands as clients of maternal care services and to understand possible complications related to pregnancy and childbirth.

Cultural norms shape the perception of individuals and subsequently engaging in positive health behaviour like male involvement in institutional delivery. This study showed that male partners who didn't fear others while accompanying their spouse to health facilities had more odds of getting involved in institutional delivery than those who fear others. This finding was supported by other studies in Africa which reported that existing cultural norms result in undesirable reactions from family and peers of male partners, which precludes them from participating in maternal processes including skilled birth care [46, 49]. It could be related to the findings of this study which reported the most frequently mentioned socio-cultural barriers for male involvement were: considering child birth as a natural phenomenon that should not be given much attention and their culture did not allow them to discuss with wife about place of delivery. Therefore, it could be due to a reason that many societies and cultures treat pregnancy and child-birth as solely a women's issue and this may have contributed to men not being invited to learn about and engage in matters related to women's and children's health [50].

Previous studies mostly focused on the positive benefits of male involvement in maternal and child health; however, the magnitude of the association is not clear. This study assesses the impact of male partner's involvement on the female partner's use of institutional delivery. Our findings showed that active participation of men appeared to have more impact on women's utilisation of institutional delivery. This study also reinforces the findings on the significant influence of education, knowledge and attitude on maternal health and socio-cultural barriers on male involvement in institutional delivery. The impact of male partner's involvement on maternal and childbirth outcomes, how men could have better involvement and the time of involvement (during pregnancy, delivery and postpartum period) merit further investigations.

The findings of this study need to be interpreted in the light of some limitations. It was difficult to explore the contextual and social factors that may limit male partner involvement as it is quantitative in design. We relied on verbal reports of fathers in relation to male involvement in institutional delivery. This has the tendency for the respondents to over report good behaviour as participants may want to provide socially desirable responses. Also, recall bias could have been present as pregnancy may not represent as much significance to the male partner as it does to the female.

## Conclusion

Male partner's involvement in institutional delivery was inadequate. This study reported a significant beneficial impact of male involvement on maternal health through improved utilisation of institutional delivery. It implies the need to shift from women-only maternal health services to 'male-friendly', couple services, and also to improve healthcare/government policies that isolate/discourage men from having active engagement in maternal health. Education, knowledge and attitude on maternal health and socio-cultural beliefs were factors associated with male partner's involvement in institutional delivery.

Therefore, maternal health interventions should target husbands as consumers of maternal health services and efforts should be made to improve awareness of the male partner's on institutional delivery through community based health education.

## Supporting information

**S1 Fig. Socio-cultural barriers for male partners' involvement in institutional delivery in Debre Tabor town, North-West Ethiopia, 2019.**
(TIF)

**S1 Table. Male partners knowledge on maternal health in Debre Tabor town, North-West Ethiopia, 2019 (n = 477).**
(PDF)

**S2 Table. Male partners attitude towards institutional delivery in Debre Tabor town, North-West Ethiopia, 2019 (n = 477).**
(PDF)

**S1 File. Questionnaire.**
(PDF)

**S2 File. Dataset.**
(SAV)

## Acknowledgments

We are very grateful to Bahir Dar University for the approval of the ethical clearance. We would also like to thank all fathers participated in this study for their commitment to responding for our interviews. Our gratitude also goes to supervisors and data collectors.

## Author Contributions

**Conceptualization:** Kassanesh Melese Tessema.

**Data curation:** Kassanesh Melese Tessema, Kebadnew Mulatu Mihirete, Endalkachew Worku Mengesha, Azezu Asres Nigussie, Awoke Giletew Wondie.

**Formal analysis:** Kassanesh Melese Tessema, Kebadnew Mulatu Mihirete, Endalkachew Worku Mengesha, Azezu Asres Nigussie, Awoke Giletew Wondie.

**Investigation:** Kassanesh Melese Tessema, Kebadnew Mulatu Mihirete.

**Methodology:** Kassanesh Melese Tessema, Kebadnew Mulatu Mihirete, Endalkachew Worku Mengesha, Azezu Asres Nigussie.

**Project administration:** Kassanesh Melese Tessema, Kebadnew Mulatu Mihirete, Endalkachew Worku Mengesha.

**Resources:** Kassanesh Melese Tessema.

**Software:** Kassanesh Melese Tessema.

**Supervision:** Kassanesh Melese Tessema.

**Validation:** Kassanesh Melese Tessema.

**Visualization:** Kassanesh Melese Tessema.

**Writing – original draft:** Kassanesh Melese Tessema.

**Writing – review & editing:** Kassanesh Melese Tessema, Kebadnew Mulatu Mihirete, Endalkachew Worku Mengesha, Azezu Asres Nigussie, Awoke Giletew Wondie.

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
