## [Decision Letter · Decision Letter 0]

29 Jan 2021

PONE-D-20-32675

Male partners’ involvement in promoting institutional delivery and its effect on the actual place of delivery in Debre Tabor town, North West Ethiopia: community based survey

PLOS ONE

Dear Dr. Wondie,

Thank you for submitting your manuscript to PLOS ONE. After careful consideration, we feel that it has merit but does not fully meet PLOS ONE’s publication criteria as it currently stands. Therefore, we invite you to submit a revised version of the manuscript that addresses the points raised during the review process.

We look forward to receiving your revised manuscript.

Kind regards,

Sara Ornaghi, M.D., Ph.D.

Academic Editor

PLOS ONE

Journal Requirements:

3. In the Methods, please discuss how the questionnaire was validated and pre-tested. If these did not occur, please provide the rationale for not doing so.

4.In your discussions and conclusions please take care to avoid statements implying causality from correlational research. For example, avoid the use of terms such as "predictors/ predictions" or “effect” or “resulted in." Instead consistently use terms such as "associated with" or "associations.

5.Please upload a new copy of Figure 1 as the detail is not clear. Please follow the link for more information: https://blogs.plos.org/plos/2019/06/looking-good-tips-for-creating-your-plos-figures-graphics/" https://blogs.plos.org/plos/2019/06/looking-good-tips-for-creating-your-plos-figures-graphics/

Reviewers' comments:

Reviewer's Responses to Questions

**Comments to the Author**

1. Is the manuscript technically sound, and do the data support the conclusions?

Reviewer #1: Partly

Reviewer #2: Yes

Reviewer #3: Yes

2. Has the statistical analysis been performed appropriately and rigorously? 

Reviewer #1: Yes

Reviewer #2: Yes

Reviewer #3: Yes

3. Have the authors made all data underlying the findings in their manuscript fully available?

Reviewer #1: No

Reviewer #2: Yes

Reviewer #3: Yes

4. Is the manuscript presented in an intelligible fashion and written in standard English?

Reviewer #1: Yes

Reviewer #2: No

Reviewer #3: No

5. Review Comments to the Author

Reviewer #1: Thank you for the opportunity to review this manuscript

An interesting study drawing on the views of local participants to understand why men do not regularly participant in MCH care. It's strengths are a clear presentation of findings, however it lacks clarity and detail in relation to the rationale for the study, methods used and implications of the findings. There are minor grammatical errors and typos throughout.

Introduction:

More detail needed on the possible impact of increased male partner involvement. Are there studies which show the impact that increased male involvement can have? What does male involvement look like (e.g. attending appointments, births etc)?

The necessity of the study is in question. Still feel like the background does not provide for a clear problem statement. Together, I would suggest the authors during their write up to consider using recent references especially those from countries with similar maternal mortality rates to Ethiopia.

Why is a study in Debre Tabor town important? Can the authors tell us about maternal and child morbidity/mortality rates there? How might the findings be relevant to the international audience? This is referred to later but a line or two at the end of the introduction would clarify the importance of the study and allow the reader to determine whether the paper is relevant to them.

Methods:

Please clarify how was male involvement defined?

The description of the outcome variable is not clear. It would be important to clarify that the variable was computed based on responses to [number of items] items: as listed and having at least 6 times [provide a comprehensive list]. It would also be important to clearly state that men who responded yes to n or more items were classified as being “? involved” while those who respondent yes to n-1 or fewer items were classified as being “less ? involved”. If there is a citation to support the coding of the variable in this way, it should be provided.

Results:

Needs to be re-written as it's extremely long, check language issues in the tables as well as avoid replicating results into the discussion section. Revisions In the tables, the total N should be included in each table. In all tables, are multiple responses included? Some editing is required

Discussion

Their interpretation of the implications of this finding is also flawed – the discussion seems a bit repetitive with more focus on restating the findings rather than discussing implications for the findings or possible future directions related research could take based on the findings. If the authors can be clearer about the added value of their study, or the importance for their setting. It would help to have a clear idea of the purpose behind the current objectives. The similarities and differences of this study needs to be brought out more clearly so that the reader can see how it adds to the current evidence. The discussion section should also include methodological issues or limitations.

The study does not give clear implications in the conclusion or recommendations. If the study provides similar findings as other studies, where is the problem? What is the explanation for this? What should be done about this situation? What should be the next logical step to address this issue? Recommendations for practice and future research should be included.

I noted that you used different reference styles so kindly use only PLOS ONE reference style through out the entire manuscript

Please provide a copy of the interview questionnaire used in this study as an additional file.

Reviewer #2: This paper seems to be a fair summary of literature on Male partners’ involvement in promoting institutional delivery and its effect on the

actual place of delivery in Debre Tabor town, North West Ethiopia: community based survey Methods of identifying relevant studies for review are thoroughly explained. My only concerns are as follows;

An English editorial correction is needed

No scientific concern noted in then article.

Reviewer #3: Comments to the Author

Thank you for the opportunity to review this article, which addresses an important and under-researched topic: namely, the role that male partners play in promoting institutional delivery, and by extension, female reproductive health outcomes. The study found that in North West Ethiopia, male spouses’ involvement in promoting institutional delivery was a strong predictor of institutional delivery, and that factors such as education, maternal heath knowledge, and favorable attitudes toward institutional delivery and spousal support were significantly associated with their involvement. As the authors note, many studies on maternal healthcare utilization exclude men, and I find this piece novel and refreshing for its attention to men’s perspectives and experiences. That said, I have several suggestions the authors might consider in order to strengthen the manuscript.

First, I wonder if the authors could ground their study a bit more in the existing literature on the topic of male partner involvement in promoting female reproductive health. To start, it would be helpful for them to define what is mean by the term “male partner involvement” in this context. How are men expected to ensure women’s and children’s health? Does it refer to men attending ANC appointments, being present at time of delivery, arranging transport and payment of clinic fees, etc., or does “involvement” go beyond this rather narrow, biomedical framing? Second, could you be clearer as to why and how male involvement/support is so important for female reproductive health? You mention the role of men in decision-making regarding healthcare seeking, but could you say more? The article you cite stating the lack of evidence on this theme is from 2012 and I think you’ll see that since this time, quite a bit has been published on the topic of male partner involvement, including a number of intervention studies and those assessing factors that either encourage/discourage men from becoming involved in this arena. I would also recommend that you contextualize the issue within Ethiopia a bit more. What is the burden of maternal and child morbidity/mortality in this context? What factors influence poor reproductive health outcomes, including low rates of institutional delivery?

As for your methods section, there are some important details that are missing and/or unclear. For instance, in your abstract you mention that the study included married men who have children less than one year of age, but in the Population section of your Methods, you just indicate “all male partners.” How are you defining “male partner”? This is an important distinction to make, as I would imagine marital status would influence the extent of men’s involvement/influence on women’s reproductive health. Additionally, I am wondering why you chose only to include men with children one year of age and below? Does this have to do with recall bias?

Where were data collected – i.e. were participants interviewed in their homes or within a health facility? What was the specific random sampling technique utilized (how did you choose where to start) and what kind of systematic technique was used thereafter? When you mention that data were collected using a questionnaire adopted from previous studies (pg. 6, line 79) can you cite these previous studies or say more about them? Also, how were they adapted? Did adaptation go beyond simple translation, and if so, how?

I also think there is much room for your discussion section to be strengthened. As currently written, it contains much repetition of your findings, and also mostly focuses on how your findings are similar to those of other studies. I would like to see you emphasize more that which differentiates your study. What new knowledge does it add to the literature? And going a step further, I think you could speak more specifically to the implications of your findings for policy and programming. Clearly, you can conclude that interventions promoting greater institutional delivery would do well to involve men, but you could also speak to why and how men could be better involved. Your findings regarding joint decision making by couples seem really important here; perhaps interventions should strive to be more gender inclusive? Further, how might policies be enacted to help address some of the barriers to men’s participation that you’ve identified in this study? Do you think other barriers – beyond knowledge and attitude – might need to be addressed? What about time and cost-related factors that may prevent men from becoming more involved in the biomedical encounter? Again, I think this comes back to the question of defining what “involvement” means, as I mentioned earlier.

On a smaller note, you say that men’s involvement in maternal health is “culturally discouraged and remains an important barrier” (pg. 17, line 246) but then you cite a study from Nepal. Can you speak to what current literature from sub-Saharan Africa suggests, and particularly those studies conducted in Ethiopia? Why is men’s participation socially discouraged in this context? Around line 282 you mention men may fear “being ruled” by their wives. Where does this explanation come from, can you cite this? Further, I would encourage you to consider how gender norms are not static but constantly changing. Is there any evidence to suggest that gender norms around male participation in the female reproductive sphere are shifting? And if so, in light of what factors?

Finally, there are a substantial number of issues relating to grammar or awkward wording that require editing. I would highly recommend careful editing during the revision process, perhaps in conjunction with a professional copyeditor. This is needed, as it’s disrupting the flow of what is otherwise an important study.

Minor comments

Pg. 3, line 2 (Introduction) – the first sentence does not make grammatical sense. Try “in which many women are exposed”

The next two sentences of the Introduction need citations.

Pg. 3, line 10 – I would specify that you mean 85% of *maternal deaths

Pg. 7, lines 95-97: I would recommend referring your readers to Table 4 here in case they are wondering how you’re measuring “knowledge about maternal health.” The same goes for “partner attitude” with respect to Table 5.

Pg. 8, line 124: Could you contextualize respondents’ average monthly income for those of us unfamiliar with Ethiopian currency? Are participants able to meet their daily needs with this level of income? How does it compare to the rest of the country?

6. PLOS authors have the option to publish the peer review history of their article (what does this mean?). If published, this will include your full peer review and any attached files.

Reviewer #1: **Yes: **Richard KALISA MD, PhD

Reviewer #2: **Yes: **Brian Barasa Masaba

Reviewer #3: No

---

## [Author Response · Author response to Decision Letter 0]

12 Mar 2021

All comments raised by academic editor and reviewers were incorporated.

---

## [Editor Report · Decision Letter 1]

29 Mar 2021

The association between male involvement in institutional delivery and women's use of institutional delivery in Debre Tabor town, North West Ethiopia: community based survey

PONE-D-20-32675R1

Dear Dr. Wondie,

We’re pleased to inform you that your manuscript has been judged scientifically suitable for publication and will be formally accepted for publication once it meets all outstanding technical requirements.

Kind regards,

Sara Ornaghi, M.D., Ph.D.

Academic Editor

PLOS ONE
---

## [Editor Report · Acceptance letter]

1 Apr 2021

PONE-D-20-32675R1 

The association between male involvement in institutional delivery and women’s use of institutional delivery in Debre Tabor town, North West Ethiopia: community based survey 

Dear Dr. Wondie:

I'm pleased to inform you that your manuscript has been deemed suitable for publication in PLOS ONE. Congratulations! Your manuscript is now with our production department. 

Kind regards, 

on behalf of

Dr. Sara Ornaghi 

Academic Editor

PLOS ONE